# Asymmetric synthesis of stereogenic-at-sulfur compounds via biocatalytic oxidation with Unspecific Peroxygenases

Jiacheng Li[1,4], Benjamin Melling[1,4], Katy A. S. Cornish [1,2], Nicholas Mulholland[3], Jared Cartwright[2], William P. Unsworth [1] ✉ & Gideon Grogan [1] ✉

Stereogenic-at-sulfur compounds are vitally important biologically active small molecules, with many drugs featuring chiral sulfur atoms. Methods for the asymmetric synthesis of sulfoxide centres are well established, but methods that produce enantiomerically enriched sulfoximines and sulfinimines are far less well developed, with no known biocatalytic methods based on oxygenation. In this study, we demonstrate that Unspecific Peroxygenases (UPOs) catalyse the biocatalytic oxygenation of sulfilimines and sulfenimines to form enantiomerically enriched sulfoximines and sulfinimines respectively, on preparative scale. In the sulfilimine series, sulfoximines are generated in up to 98% *ee* via a kinetic resolution approach. In the sulfenimine series, the selective, synthesis of both (*R*)- and (*S*)-sulfinimine products (both up to 99% *ee*) can be achieved, with different UPOs affording products with opposite enantioselectivity. Both series represent additional applications of UPO technology to an ever-growing list of selective, practical and industrially relevant biotransformations.

Enantioenriched stereogenic-at-sulfur compounds have myriad important roles, spanning medicinal chemistry[1–5], agrochemistry[6] and asymmetric synthesis/catalysis (Fig. 1a)[7–9]. For example, the blockbuster gastro-intestinal drug Esomeprazole - prepared as its (*S*)-enantiomer **1a** – has improved metabolic stability and inhibits gastric acid production more effectively than Omeprazole, the racemic variant that preceded it[10]. Both enantiomers of *tert*-butanesulfinamide **1b** (often referred to as Ellman's sulfonamide) are available commercially, and are highly effective chiral auxiliaries used in the asymmetric synthesis of amine derivatives[11–14]. Enantipure sulfur compounds are also important in asymmetric organocatalysis (e.g., **1c**)[7–9]. In terms of biological applications, sulfoximines are arguably the most important class of sulfur(IV) compound, featuring in several commercial products, such as the insecticide Sulfoxaflor **1d**[6]. They are also of major current interest in medicinal chemistry (e.g., anti-cancer drug **1e**)[15]. Sulfoximines often exhibit good pharmacokinetic properties, such as

high metabolic stability and solubility, rendering them useful isosteres of sulfones and sulfonamides[1–5,15]. The additional nitrogen site opens the avenue for H-bonding interactions, and enables easy manipulation of the physiochemical properties through *N*-functionalisation reactions.

Increased interest in chiral sulfur(IV) compounds has helped to propagate a significant upsurge in the development of methods for their synthesis. This includes catalytic asymmetric methods, often based on kinetic resolution strategies, transition metal catalysis, or methods using organocatalysts[16–24]. A notable recent approach, developed by Tian, Xie, Guo, and coworkers and summarised in Fig. 1b[25], is their efficient, enantioselective method for the catalytic asymmetric synthesis of chiral sulfinamides **2a** and sulfinate esters **2b**, enabled by bifunctional 4-arylpyridine *N*-oxide organocatalysts.

Biocatalytic methods to prepare stereogenic-at-sulfur compounds are far less well-developed in comparison, with methods based

[1]Department of Chemistry, University of York, York, UK. [2]Department of Biology, University of York, York, UK. [3]Jealott's Hill International Research Centre, Syngenta, Berkshire, UK. [4]These authors contributed equally: Jiacheng Li, Benjamin Melling. ✉e-mail: william.unsworth@york.ac.uk; gideon.grogan@york.ac.uk

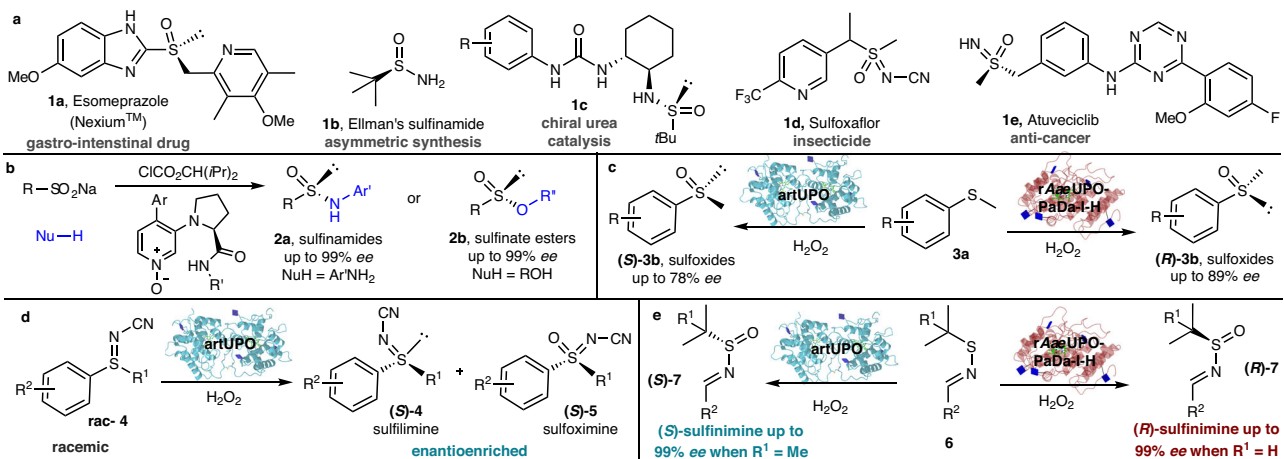

**Fig. 1 | Stereogenic-at-sulfur compounds and methods for their synthesis.**
**a** Examples of stereogenic-at-sulfur compounds used as pharmaceuticals, agro-chemicals and in asymmetric synthesis/catalysis. **b** Wei and coworkers[25]: enantio-selective synthesis of sulfinamides and sulfinate esters using bifunctional 4-arylpyridine N-oxide organocatalysts. **c** Robinson and coworkers[33]: sulfide to sulfoxide oxygenation using a UPO. **d** This work: enantioselective sulfoximine synthesis via the UPO catalysed oxygenation of sulfilimines, using a kinetic reso-lution approach. **e** This work: enantioselective synthesis of sulfinimine enantiomers via the UPO catalysed oxygenation of sulfenimines.

on oxygenation limited to the conversion of sulfides into sulfoxides[26]. A range of oxygenase enzymes, including hemoproteins such as cytochromes P450[27,28] and flavoprotein oxygenases, such as Baeyer-Villiger Monooxygenases (BVMOs)[29,30], has been shown to catalyse this transformation, with one example reported to work on a kg scale for the production of a pharmaceutical intermediate by AstraZeneca[31]. Among hemoprotein biocatalysts, the conversion of simple sulfides into sulfoxides has also been reported for Unspecific Peroxygenases (UPOs)[32,33]. UPOs have been the focus of substantial recent research[34,35], as they display advantages over P450s and BVMOs in that they are nicotinamide cofactor independent and require only the addition of hydrogen peroxide to promote their oxygenation reac-tions. In addition to these advantages, a substantial number of enzymes has now been identified[36,37], and these have been shown to catalyse the selective oxidation of carbon atoms in a range of sub-strates, including simple alkanes[38], fatty acids[39], aromatics[40], terpenes[41,42] and pharmaceuticals[43], as well as being capable of pro-miscuous activities, such as halogenation[44] and cyclopropanation[45]. UPOs have been classified into two broad subclasses, Class I and Class II, based on characteristics including sequence, structure, and mole-cular weight[46]. Class I UPOs are smaller, with MWs of around 29 kDa and Class II UPOs are larger enzymes, with MWs of around 44 kDa. Class I and Class II often promote biotransformations with different selectivities, and indeed, in previous work[33], we showed that a Class I UPO, an engineered artificial peroxygenase[47] (artUPO), related to the enzyme from *Marasmius rotula*[48], converts phenyl methyl sulfides **3a** into (S)-sulfoxide products (S)−**3b**, thus displaying complementary enantioselectivity to the Class II *Aae*UPO, which generates (R)−**3b** (Fig. 1c)[32,33]. This divergent reactivity clearly has great potential for exploitation with respect to the oxidation of more complex substrates, such as the biocatalytic oxidation reactions to form S(IV) products that are the subject of this study.

In this manuscript, we describe methods for the biocatalytic oxygenation of two distinct families of sulfur precursors (Fig. 1d and e). These methods allow the enantioselective synthesis of sulfoximines (Fig. 1d) and sulfinimines (Fig. 1e) on preparative scale, using two easy-to-handle UPO enzymes. The sulfoximine-forming series relies on a UPO-mediated sulfilimine kinetic resolution, while in the sulfinimine-forming series (Fig. 1e), different UPOs are able to selectively deliver either enantiomer ((S)−**7** or (R)−**7**) of the sulfinimine product with high enantioselectivity, via UPO catalysed sulfenimine oxygenation. Both approaches enable the biocatalytic formation of important

stereogenic-at-sulfur(IV) compounds to be prepared in high *ee* on preparative scale. These approaches all represent valuable additions to the biocatalytic toolbox, and further expand the range and utility of UPOs to perform useful and selective preparative biocatalytic oxyge-nation reactions.

## Results and discussion

Given their importance in medicinal chemistry, we started by explor-ing the potential of UPOs to generate enantioenriched sulfoximines. Various powerful chemical methods for the asymmetric synthesis of sulfoximines and their derivatives have been developed in recent years; these include methods based on the chromatographic separa-tion and resolutions of racemic sulfoximines[49–52], the oxidative imination of sulfoxides[53,54], electrophilic addition reactions to S-nucleophiles[51,55–57] and nucleophilic addition reactions to S-electrophiles[51,58–62]. An alternative approach for sulfoximine synthesis is via the oxidation of a sulfilimine (**4 → 5**). Methods for the oxidation of sulfilimines (also known as sulfimides) to form racemic sulfoximines have been known for decades, typically using simple peroxide based chemical oxidants[63–65]. However, to the best of our knowledge, all published syntheses of enantioenriched sulfoximines from sulfilimines rely on the stereospecific oxidation of an enantiomerically enriched sulfilimine precursors[16], with the requisite sulfilimine starting material typically prepared from sulfides, e.g., using an asymmetric transition metal catalysed nitrene transfer reaction[66,67]. Methods for the pre-paration of enantiomerically enriched sulfilimines are rare, with nota-ble exceptions being those noted above[66,67], a biocatalytic (cytochrome P450$_{BM3}$) approach from Farwell and coworkers[68], and a recently reported organocatalytic method by Wang and coworkers[69].

In recognition of the well-established ability of UPOs to catalyse enantioselective oxygenation reactions, we questioned whether they might be capable of forming enantioenriched sulfoximines via the kinetic resolution of easy-to-prepare racemic sulfilimines. To the best of our knowledge, no asymmetric sulfilimine to sulfoximine transfor-mations are known that start from racemic starting materials. Fur-thermore, we also know of no biocatalytic methods (neither racemic nor asymmetric) for the oxidation of sulfilimines, using UPOs or indeed any other enzyme class.

In view of their stability and ease of synthesis, N-cyano sulfilimines of the type **rac-4** were selected for this study, with a view towards developing biocatalytic kinetic resolution transformations of the type summarised in Fig. 2 (**rac-4 → (S)−4** and **(S)−5**). We started by exploring

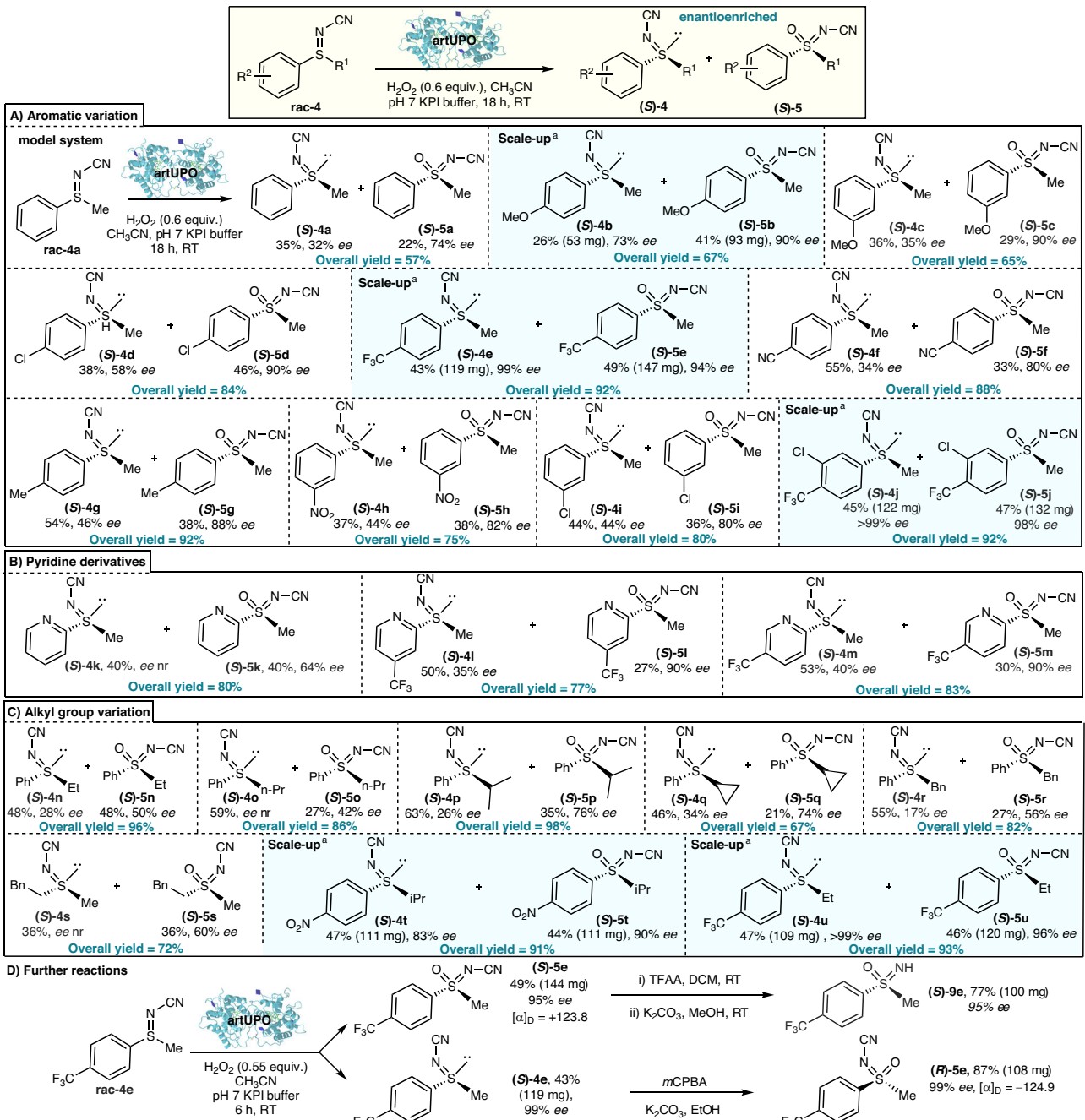

**Fig. 2 | Enantioselective sulfoximine synthesis via the UPO catalysed oxygenation of sulfilimines, using a kinetic resolution approach. A** Aromatic variation; **B** Pyridine derivatives; **C** Alkyl group variation; **D** Further reactions. Unless stated, the following reaction conditions were used: mix KPi pH 7 buffer (24 mL), CH₃CN (6 mL), artUPO secrecate (1 mL, 0.8 U/mL) and **4** (0.3 mmol), then H₂O₂ (0.180 mmol) added over 4 h, followed by stirring overnight at RT. All % yields refer to pure product, isolated by column chromatography, in a preparative scale biotransformation. ᵃ scale up reactions performed using 1 mmol **4** with additional H₂O₂ additions used until ≈50% conversion was observed by ¹H NMR analysis (see Supplementary Information section 2.5).

the kinetic resolution of the simple *N*-cyano phenyl methyl sulfilimine **rac-4a**, which was prepared via a straightforward oxidative procedure from phenyl methyl sulfide[65]. All other racemic sulfilimine precursors used throughout this study were made using similar methods (see Supplementary Information Sections 2 and 3 for full preparative details and characterisation data). Our aim was to develop a kinetic resolution of racemic sulfilimine starting material **rac-4a** and to isolate the sulfoximine product **5a**, along with unreacted sulfilimine **4a**, both in enantioenriched form.

The biotransformation of **rac-4a** was therefore tested conducted on a preparative (0.3 mmol) scale using artUPO and the conditions

summarised in Fig. 2; the use of 0.6 equivalents of H₂O₂ was chosen to facilitate approximately 50% conversion of **rac-4**, as needed for effective kinetic resolution (for optimisation details, control experiments and a time-course experiment, see Supplementary Information, Sections 2.5, 2.6 and Figure S1). Encouragingly, under these conditions kinetic resolution was achieved, with enantioenriched sulfoximine **(S)-5a** and sulfilimine **(S)-4** isolated in 22% and 35% yields, and 74% and 32% *ee* respectively following column chromatography. The absolute configuration of the major enantiomer **(S)-5a** was confirmed by comparison to literature optical rotation data[64]; therefore, the absolute configuration of the major enantiomer of the unreacted sulfilimine is

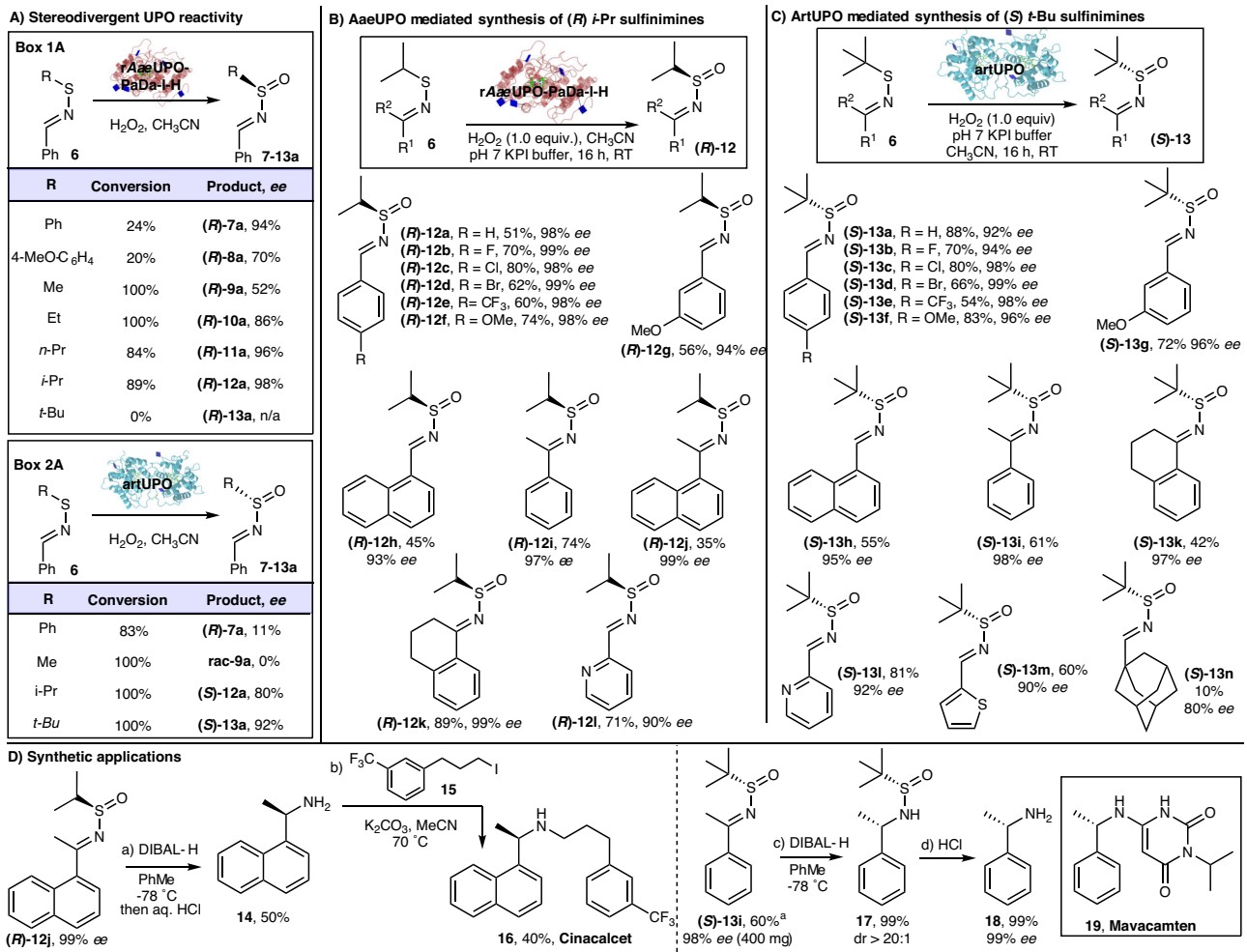

**Fig. 3 | Enantioselective synthesis of both sulfinimine enantiomers via UPO catalysed oxygenation of sulfenimines. A** Stereodivergent UPO reactivity; **B** AaeUPO mediated synhesis of (*R*)- *i*-Pr sulfinimines; **C** ArtUPO mediated synthesis of (*S*)-*t*-Bu sulfinimines; **D** Synthetic Applications. Unless stated, the following reaction conditions were used: KPi pH 7 buffer (10 mL), r*Aae*UPO-PaDa-I-H (1.3 mL, 57 U/mL) or artUPO (1.0 mL, 0.8 U/mL) and **6** (0.2 mmol), H₂O₂ (0.2 mmol) added over 10 h, followed by stirring for a further 6 h at RT. All % yields refer to pure product, isolated by column chromatography, in a preparative scale biotransformation. ᵃ scale up reaction performed using 3 mmol of **6** (see Supplementary Information for full details).

logically assumed to be (**S**)−**4**. Note that while products (**S**)−**5a** and sulfilimine (**S**)−**4** both have (**S**)-stereochemical assignments, this merely reflects the change in Cahn–Ingold–Prelog priorities, and they have opposite sense of absolute stereochemistry as expected. The assignment of absolute configuration of the other related products in this manuscript was made by analogy, supported by optical rotation and HPLC data (see Supplementary Information, Section 3 and 5). The Class I enzyme *Dca*UPO from *Daldinia caldariorum*[70] and the Class II UPOs r*Aae*UPO-PaDa-I-H[71] and *Cma*UPO from *Coprinopsis marcescibilis*[72] were also tested as alternative UPOs for this transformation but were not pursued, as they led to much poorer conversions or enantioselectivity when tested on sulfilimine substrates under the same conditions (see Supplementary Information, Section 2.6).

Having established that the proposed UPO mediated kinetic resolution is viable, attention turned to exploring the scope of the artUPO biotransformation with other sulfilimine substrates (Fig. 2A–C). All biotransformations were performed on preparative scale, with the yields quoted referring to purified products following column chromatography. The *ee*s of the product sulfoximines (**S**)−**5** were measured using chiral HPLC analysis of the isolated product; the *ee*s of the enantioenriched sulfilimines (**S**)−**4** were also measured by chiral HPLC, in some cases from the sulfilimines directly, and on others following *m*-CPBA oxidation to the corresponding sulfoximines when

this expedited analysis (see Supplementary Information, General Procedure 2.3).

We started by exploring substrates similar to **rac-4a** with different substituents on the phenyl ring (Fig. 2A). Ten substrates of this type were tested, with successful kinetic resolution achieved and enantioenriched sulfoximines ((**S**)−**5a**−**k**) and sulfilimines ((**S**)−**4a**−**k**) isolated in all cases. Electron-rich and -poor substituents were both well tolerated. Substrates featuring a *para*-trifluoromethyl substituent (**rac-4e** and **rac-4j**) worked especially well, with very efficient kinetic resolution achieved, with >90% overall yield and >95% *ee* obtained for the isolated sulfoximines ((**S**)−**5e, 5j**) and enriched sulfilimines ((**S**)−**4e, 4j**). Three pyridine-containing substrates (Fig. 2B) were also transformed successfully, with good to high *ee*s for the sulfoximine products ((**S**)−**5k**−**m**) obtained in this series. It is notable that the protocol can be applied to aza-heterocyclic systems given their prominence in medicinal chemistry, while avoiding the proclivity of pyridines to undergo oxidation to form *N*-oxides with UPOs[73]. Fig. 2C summarises results using racemic sulfilimine starting materials with non-methyl alkyl groups (**rac-4n**−**u**). The resolution of substrate **rac-4s** is notable, as this shows that the biotransformation is not contingent on the sulfilimine having an aromatic substituent. As before, the most effective examples in this series were those containing electron-poor aromatic groups, with sulfilimines **rac-4t** and **rac-4u** affording products in >90%

overall yield, and with *ee*s of 90% and 96% respectively for sulfoximines **(S)-5t** and **(S)-5u**.

Except for selected cases (those highlighted with a blue background) the biotransformations were all performed on 0.3 mmol scale using the same conditions, without optimisation and on a substrate-by-substrate basis. Of course, for any kinetic resolution, it is unlikely that the same reaction conditions will be optimal across all substrates. To demonstrate how additional optimisation can lead to improved results, and to showcase the scalability of the method, the biotransformations of **rac-4b, rac-4e, rac-4j, rac-4t** and **rac-4u** were performed on larger (1.0 mmol) scale, leading to the improved results presented (Fig. 2A and C, examples highlighted with a blue background). The increase in scale was useful in facilitating more precise control in reaction conversion compared with the standard method; careful monitoring of conversion and the use of additional $H_2O_2$ when needed, proved to be effective in achieving the ≈50% conversion needed for optimal kinetic resolution (see Supplementary Information, General Procedure 2.5). The resolution of substrate **rac-4j** (98% and >99% *ee*, s > 200) best illustrates the power of this approach.

*N*-Cyano sulfilimines were chosen as starting materials for this study primarily in view of their synthetic tractability. However, if the unfunctionalised sulfoximine products are required, either directly or for further derivatisation, these can easily be accessed using the method summarised in Fig. 2D[74]. Thus, reaction of *N*-cyano sulfoximine **(S)-5e** with trifluoroacetic anhydride followed by potassium carbonate in methanol affords sulfoximine **(S)-9e** with no erosion in *ee*. Furthermore, if the enantiomeric **(R)**-sulfoximine is required, this can be obtained via simple oxidation of the enantiomerically enriched **(S)**-sulfilimine; for example, the reaction of sulfilimine **(S)-4e** with *m*-CPBA gave sulfoximine **(R)-5e** in good yield, and with no erosion in *ee*. Note that the oxidation of **(S)-4e** into **(R)-5e** is stereo-retentive, and the change from an **(S)**- to **(R)**-stereochemical assignment merely reflects the change in Cahn–Ingold–Prelog priorities.

The enantiopreference of artUPO for *N*-cyano sulfilimines was investigated through docking of the favoured **(R)**-enantiomer of **4e** into our previously obtained structure of artUPO (PDB 7ZNM)[33] using Autodock VINA[75]. The lowest energy pose obtained positions the sulphur lone pair of the *pro*-**(S)** face of **(R)-4e** ideally for receiving oxygen from the oxygenating species Compound I (CpdI) with the aromatic group bound in a hydrophobic pocket formed by L65, V69, I91, I160 and I235 (Fig. S71A) and the *N*-cyano group positioned between the side chains of I62, A66 and F167. A similar pose for the unfavoured **(S)**-enantiomer of **4e** would bring the cyano group into close contact with the side chains of I160, L163 and E164, thus providing a plausible structural explanation for the experimentally observed enantioselectivity. The lower conversion observed for *meta*-substituted substrate **4c** (29%) *versus* its *para*-isomer **4b** (41%) may be attributable to unfavourable clashes of the *meta*-substituent with the side chains of V69 or I91.

After establishing artUPO as a selective biocatalyst for the kinetic resolution of sulfilimines **4**, attention then turned to the exploration of UPOs for the enantioselective synthesis of sulfinimines. Sulfinimines (also known as *N*-sulfinylimines) are extremely useful synthetic intermediates owing to their ability to undergo a range of diastereoselective transformations, and hence have been widely used to prepare chiral amine derivatives[11–14,76]. When enantiomerically enriched sulfinimines are used in organic synthesis, most commonly they are prepared via the condensation of an enantiomerically enriched sulfinamide (often Ellman's sulfinamide auxiliary **1b**, or the *p*-tolyl derivative popularised by Davis and coworkers[77,78]) with an aldehyde or ketone. In this work, we explored an alternative approach to access enantiomerically enriched sulfinimines, via the UPO catalysed oxidation of sulfenimines **6**.

Representatives of both Class I and Class II UPOs were tested for this transformation, revealing intriguing divergent reaction profiles for the different UPO classes. First, Class II r*Aae*UPO-PaDa-I-H was challenged with a selection of seven different phenyl sulfenimines with different *S*-alkyl/aryl substituents (to form **(R)-7a–13a**, Fig. 3, Box 1 A). The reactions were performed on preparative scale (0.2 mmol of **6**) in pH 7 KPi buffer with acetonitrile as co-solvent and using 1 equivalent of $H_2O_2$ (added slowly over 10 h) as the stoichiometric oxidant (see Fig. S2 for a time-course experiment). Conversion was measured by comparing the amounts of **6** and product in the $^1$H NMR spectra of the crude reaction mixture, and *ee* was measured using chiral HPLC (see Supplementary Information, Section 5). In all but one case, some conversion into the expected enantiomerically enriched sulfinimine was observed and the depicted **(R)**-enantiomer was formed in excess; the assignment of absolute stereochemistry made by comparison to literature optical rotation data for **(R)-12a**[79] (corroborated by several other substrates in the series featured later in Fig. 3B, see Supplementary Information Section 3). The most successful example was the oxygenation of the *i*-Pr-substituted sulfenimine, which was converted into sulfinimine **(R)-12a** with 89% conversion, corresponding to a total turnover number (TTN, expressed as μmol product/μmol enzyme) of $1.57 \times 10^4$, and with 98% *ee*. The only substrate in this series that did not undergo oxygenation was the *tert*-butyl substituted sulfenimine, where no conversion into **(R)-13a** was observed. This result was not wholly surprising, given that r*Aae*UPO-PaDa-I-H tends to perform less well in the biotransformations of more sterically demanding substrates[41]. Interestingly, the enantiopreference for substrate **6** was conserved for another Class II UPO, the *Cma*UPO from *Coprinopsis marcescibilis*[72], although with reduced selectivity (see SI sections 2.6 and 4).

In contrast, artUPO, the same Class I UPO employed in the first half of this manuscript, typically performs better than r*Aae*UPO-PaDa-I-H with bulky substrates, probably owing to its more accessible active site[33]. Thus, four sulfenimines were tested using artUPO in the same way (Fig. 3, Box 2A). All four substrates were converted well, included the bulky *tert*-butyl substituted sulfenimine, with full conversion into **(S)-13a** observed, corresponding to a TTN of $5.88 \times 10^3$ (see SI, Fig. S3 for a time-course experiment). For the smaller substrates, the products **(R)-7a** and **rac-9a** were formed with little or no *ee* respectively; again, this was not wholly surprising, as the ability to transform bulkier substrates using artUPO is often offset against reduced enantioselectivity. However, for the bulkier substrates, enantioselectivity was much higher, with products **(S)-12a** and **(S)-13a** formed in 80% and 92% *ee* respectively. Remarkably, the opposite **(S)**-enantiomer was formed in excess in this series, thus offering complementary enantioselectivity to that of the Class II enzyme r*Aae*UPO-PaDa-I-H. The enantiopreference of artUPO was also conserved for an additional Class I UPO, *Dca*UPO from *Daldinia caldariorum* (see SI sections 2.6 and 4)[70]. The assignment of absolute stereochemistry was made by comparison to literature optical rotation data for **(S)-13a**[80], and corroborated in several cases in the series in Fig. 3C (see Supplementary Information Section 3). The complementary enantioselectivity accords with previous preliminary observations made for r*Aae*UPO-PaDa-I-H and artUPO for the oxidation of sulfides[33]. In the case of sulfenimines such as **6**, this complementarity appears to result from different substrate approach trajectories to the oxidising species CpdI, in the active sites of the enzymes, as revealed by docking **6** (where R = *i*-Pr) into the enzymes, again using Autodock VINA (see Supplementary Information, Fig. S71B)[73]. Hence in the case of artUPO, the *pro*-**(S)** lone pair is presented to the CpdI oxygen as the phenyl ring rests in the hydrophobic pocket formed by L65, V69, I91, I160 and I235 previously described. As the equivalent pocket in r*Aae*UPO-PaDa-I-H is restricted by phenylalanine residues including F188, the substrate approaches CpdI through a more available and less constrained tunnel and thus presents the *pro*-**(R)**-lone pair to the oxidant.

The best performing cases with each enzyme (*i*-Pr and *t*-Bu derivatives, leading to the formation of **(R)-12a** and **(S)-13a**) are arguably amongst the most synthetically useful systems, considering the

established utility of *i*-Pr and *t*-Bu sulfinimines in asymmetric synthesis[11–14]. Therefore, we next examined the scope of these biotransformations on other *iso*-propyl- and *tert*-butyl-substituted sulfenimines. Results for the enantioselective synthesis of *iso*-propyl substituted (*R*)-sulfinimines **7** using r*Aae*UPO-PaDa-I-H are summarised in Fig. 3B. As in the initial screening, the reactions were performed on preparative scale (0.2 mmol of **6**) in pH 7 KPi buffer with acetonitrile as co-solvent and using 1 equivalent of $H_2O_2$ (added slowly over 10 h) as the stoichiometric oxidant. The yields quoted refer to isolated yields of purified products following column chromatography, and *ee* was measured by chiral HPLC. All twelve examples proceeded with excellent enantioselectivity (90–99% *ee*), across a range of substituted aromatic substrates ((*R*)−**12a**–**i**). Naphthyl ((*R*)−**12h**, (*R*)−**12j**), ketimine ((*R*)−**12i**–**k**), cyclic ((*R*)−**12k**) and pyridyl ((*R*)−**12l**) substrates also worked well.

The opposite enantiomeric series is depicted in Fig. 3C, in which results for the formation of *tert*-butyl substituted (*S*)-sulfinimines (*S*)−**13** using artUPO are summarised. Enantioselectivity was again high, with the (*S*)-enantiomer formed in >90% *ee* in most examples tested. As before, a range of substituted aromatic substrates were tested and all worked well ((*S*)−**13a**–**g**). Naphthyl ((*S*)−**13h**), ketimine ((*S*)−**13i**), cyclic ((*S*)−**13j**) and heterocyclic ((*S*)−**13l**–**m**) substrates were also well tolerated. The poor conversion (10% yield) for adamantyl derivative (*S*)−**13n** appears to show the limits of enzyme with respect to steric bulk of the substrate, although notably the enantioselectivity remained relatively high (80% *ee*). In both series, attempts to test simple alkyl substituted sulfenimines (e.g., **6** where neither $R^1$ nor $R^6$ is aromatic) were thwarted by the instability of the requisite starting materials, meaning the biotransformations were not tested. The method is therefore limited to aromatic and cyclic aliphatic sulfenimines systems, that can be prepared and handled easily.

To showcase the utility of the method of the enantiomerically enriched sulfinimines accessible using this method, two relatively simple syntheses of secondary amines used on the synthesis of pharmaceuticals are summarised in Fig. 3D. Starting from sulfinimine (*R*)−**12j** (prepared in 99% *ee* using r*Aae*UPO-PaDa-I-H), DIBAL reduction, followed by hydrolysis delivered a single enantiomer of amine **14**. Alkylation with iodide **15** then delivered hyperparathyroidism drug Cincalcet **16**. Similarly, in the opposite series, sulfinimine (*S*)−**13i** was prepared in 98% *ee*; in this case, the reaction was performed on 3 mmol scale and delivered 400 mg of the enantiomerically enriched product, in almost identical yield to the smaller scale version. A highly diastereoselective reduction and hydrolysis followed to deliver amine **18** in 99% *ee*, with this amine a key precursor to the obstructive hypertrophic cardiomyopathy drug Mavacamten **19**.

In summary, two preparative biocatalytic approaches for high yielding and enantioselective oxygenation to form stereogenic-at-sulfur(IV) compounds have been developed and described. To the best of our knowledge, both approaches had no known biocatalytic variants, using any enzyme, prior to this manuscript. Selective oxidation reactions in organic chemistry remain a challenge, especially where enantioselective transformations are concerned. Whereas cofactor-dependent enzymes present problems with stability, turnover, and expense, UPOs offer real potential for scalable asymmetric oxidations using simple procedures. To facilitate their wider uptake, it is crucial that examples of scalable reactions on useful molecules are investigated and presented. With the transformations to form stereogenic-at-sulfur functionalities presented in this report, we add another highly promising application of UPO technology to a growing list of reactions that have significant potential for scale-up in an industrial context.

## Methods

General procedures for used for preparative biosynthetic reactions are as follows. More details, can be found in the Supplementary Information.

### General procedure for artUPO kinetic resolution of sulfilimines (4) at 0.3 mmol scale (Fig. 2)

Liquid artUPO secretate (1.0 mL, 0.8 U/mL) was added to KPi Buffer (24.0 mL, 100 mM, pH 7.) at RT and stirred for five min, after which, a solution of sulfilimine (0.300 mmol) in MeCN (6.00 mL) was added. The reaction was initiated by the slow continuous addition of a $H_2O_2$ solution (0.180 mmol in 2 mL $H_2O$) over 4 h followed by stirring overnight. The reaction was extracted with diethyl ether (3 ×30 mL), and the combined organic phase washed with saturated brine (40 mL), dried over $MgSO_4$, filtered and the solvent removed in vacuo. For preparative reactions, the purified products were isolated following flash column chromatography on silica gel.

### General procedure for r*Aae*UPO biotransformations with *i*-Pr *N*-sulfenylimines (Fig. 3B)

To a round bottom flask containing a magnetic stirring bar was added r*Aae*UPO-PaDa-I-H (1.3 mL, 57 U/mL) and KPi buffer (10 mL, 100 mM, pH = 7). The solution was diluted by the addition of deionised water (2.7 mL), followed by addition of the appropriate *i*-Pr *N*-sulfenylimines (0.2 mmol, 1.0 equiv.,) in MeCN (4 mL). Next, $H_2O_2$ solution (2 mL, 0.1 mmol/mL, 1.0 equiv.) was added over a 10 h period, using a syringe pump. After the $H_2O_2$ addition was complete, the reaction was then stirred at room temperature for a further 6 h. The reaction mixture was then extracted with ethyl acetate (3 × 20 mL). The combined organic phases were then washed with brine (20 mL), dried over anhydrous $MgSO_4$ and concentrated *in vacuo* to give the crude product mixture, which was purified by flash column chromatography on silica to provide the corresponding *N*-sulfinyl imine product (*R*)−**12**.

### General procedure for the artUPO biotransformations with *t*-Bu *N*-sulfenylimines (Fig. 3C)

To a round bottom flask containing a magnetic stirring bar was added liquid artUPO secretate (1.0 mL, 0.8 U/mL) and KPi buffer (10 mL, 100 mM, pH = 7). The solution was diluted by the addition of deionised water (3 mL), followed by addition of the appropriate *t*-Bu *N*-sulfenylimine **6** (0.2 mmol, 1.0 equiv., final concentration 10 mM) in MeCN (4 mL). Next, 2 mL of a 100 mM $H_2O_2$ solution (prepared from 22 μL 30% $H_2O_2$ in 2 mL deionised water) was added over a 10 h period, using a syringe pump. After the $H_2O_2$ addition was complete, the reaction was then stirred at room temperature for a further 6 h. The reaction mixture was then extracted with ethyl acetate (3 × 20 mL). The combined organic phases were then washed with brine (20 mL), dried over anhydrous $MgSO_4$ and concentrated *in vacuo* to give the crude product mixture, which was purified by flash column chromatography on silica gel to provide the corresponding *N*-sulfinyl imine product (*S*)−**13**.

## Data availability

The authors declare that the data supporting the findings of this study are available within the paper and its Supplementary Information files. Should any raw data files be needed in another format they are available from the corresponding author upon request.

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

## Acknowledgements

We thank the U.K. Engineering and Physical Sciences Research Council (EPSRC, EP/X014886/1, J.L. and K.A.S.C.) for funding, and the EPSRC and Syngenta for the award of a PhD studentship to B.M. (project 2602946).

## Author contributions

W.P.U. and G.G. designed the study. J.L., B.M., and K.A.S.C. performed the experiments and interpreted the results. J.C. led on all aspects relating to enzyme production. N.M. provided industrial advice and additional guidance (to B.M.). W.P.U. and G.G. prepared the manuscript and for publication, supported by J.L. and B.M.

## Competing interests

The authors declare no competing interests.
