## [Transparent Peer Review file · Nature Communications]

Asymmetric synthesis of stereogenic-at-sulfur compounds via biocatalytic oxidation with Unspecific Peroxygenases

Corresponding Author: Professor Gideon Grogan

Version 0:

Reviewer comments:

Reviewer #1

(Remarks to the Author)

Li and colleagues designed a biocatalytic method for converting sulfur-containing substrates into chiral sulfoxides using unspecific peroxygenases. Specifically, based on the selection of different peroxygenase enzymes, biocatalytic oxygenation of sulfilimines and sulfenimines to form enantiomerically enriched sulfoximines and sulfinimines on preparative scale was achieved. The sulfoximines are generated in up to 99% ee via kinetic resolution reaction using ArtUPO, while the sulfinimines were obtained with complementary chirality using the combination of ArtUPO and AaeUPO. These two methods indeed provide new catalytic insights into the unspecific peroxygenase catalysis, and also establish new catalytic pathways for synthesizing chiral sulfur-compounds. I found this work interesting and promising, and well-designed. Prior to publishing the manuscript, some corrections are necessary to improve the work further. Details are shown below.

1. In the introduction section, it was too focused on the sulfur-containing compounds (e.g., their importance, synthesis, etc.). To broaden the interest of readership, it would be great if the authors could share some attention to the peroxygenase catalysis, particularly considering the fact that the advances of this class of enzymes are expanding fast, lots of new catalytic pathways and new products have been reported.

2. The authors state: "However, to the best of our knowledge, all published syntheses of enantioenriched sulfoximines from sulfilimines start from enantioenriched sulfilimine precursors; as sulfilimines are themselves chiral and configurationally stable, stereochemical information in the precursor is retained following oxidation". To enhance the clarity of this manuscript for readers, could the authors consider providing a summary table or a concise listing of these reported synthesis methods in the Supporting Information?

3. The description of enzymes in Tables 1 and 2 is inaccurate and lacks clarity. Please specify the amount of enzyme used in each reaction. (for example: table 1: artUPO secretate (1 mL); table 2: rAaeUPO-PaDa-I-H or art UPO and 6 (0.2 mmol))

4. The overall yield for (S)-4i+(S)-5i appears to be incorrect based on the data provided.

5. The authors utilize rAaeUPO-PaDa-I-H (Class I) and ArtUPO (Class II) as representative enzymes and demonstrate differing reactivity and selectivity patterns. To further broaden the impact of this study, have the authors considered exploring the applicability of these findings to other UPO sources? Specifically, do other Class I and Class II UPOs exhibit similar reactivity and selectivity patterns?

6. Please clearly explain the meanings of the conversions (Table 1 and Table 2) and how they were determined.

7. I would prefer to see a reaction time course of both model reactions to see the initial reaction rate and robustness of the overall reaction scheme. It would be great if the authors can provide this and place them in the Supplementary Information.

8. Line 101: Zhang, Zhang and coworkers was wrong.

9. Supporting Information, page 68: Please describe in detail the method used to determine the UPO concentration (e.g., Bradford assay, spectrophotometric method) and/or enzyme activity. It is essential for reproducibility that detailed information regarding the enzyme activity (specific activity, total units used per reaction) and enzyme preparation (source, purification stage, formulation buffer) are provided both in the main text (Experimental Section) and comprehensively in the Supplementary Information.

Reviewer #2

(Remarks to the Author)

This work presents an innovative biocatalytic oxidation strategy for the asymmetric synthesis of valuable stereogenic-at-sulfur compounds. By harnessing the power of unspecific peroxygenases (UPOs), the authors have successfully developed a kinetic resolution process for sulfilimines to access sulfoximines with up to 99% ee, and a stereodivergent oxygenation of sulfenimines to furnish either enantiomer of sulfinimines with high enantioselectivity. The substrate scope is impressively broad, encompassing a diverse range of sulfilimine and sulfenimine derivatives with generally good yields and high enantioselectivities. Given that such efficient and selective biocatalytic methods for constructing these important chiral sulfur(IV) centers were previously unknown, this study represents a landmark contribution that significantly enriches the synthetic toolbox for asymmetric sulfur chemistry and expands the synthetic applicability of UPOs. The work is therefore suitable for publication in Nature Communications following appropriate revisions to address the points outlined below.

1. The introduction provides a compelling overview of the importance of stereogenic sulfur(IV) compounds. To further strengthen the manuscript, the organization and literature coverage should be enhanced. For example, the extensive background on chemical methods for sulfoximine synthesis in the first paragraph of the Results and Discussion would be more appropriately placed in the introduction. Furthermore, while kinetic resolution and organocatalysis are mentioned, the introduction would benefit from acknowledging other catalytic strategies, such as transition metal catalysis (e.g., *J. Am. Chem. Soc.* 2025, 147, 2137–2147; *ACS Catal.* 2025, 15, 5511–5530, *Nat. Commun.* 2025, 16, 2310; *JACS Au* 2023, 3, 700–714). Additionally, presenting 'kinetic resolution' and 'organocatalysis' as parallel categories is conceptually imprecise.
2. For the model substrate rac-4 in the sulfilimine series, the initial results, while promising, were moderate ((S)-4a: 35% yield, 32% ee; (S)-5a: 22% yield, 74% ee). It is unclear whether systematic reaction optimization—such as varying enzyme loading, H₂O₂ concentration, temperature, or reaction time—was performed for this model reaction. Including these details or comments on optimization efforts in the Supporting Information would be valuable.
3. The assignment of absolute configuration for certain compounds requires further verification. The determination for (S)-5a, based on comparison of optical rotation with a literature value obtained for a sample of significantly higher enantiopurity (74% ee vs. 99% ee), could be strengthened. Synthesizing an authentic reference standard for direct comparison via chiral HPLC would provide more definitive proof. Similarly, the assignment of (R)-12a also relies on comparison of optical rotation with literature data (ref. 30). However, the cited reference does not contain the relevant data for (R)-12a. Verification of this citation or additional evidence for configuration assignment is recommended.
4. The use of enzyme secretate is practical, but it raises the question of whether other components in the secretate could contribute to or interfere the catalysis. Additionally, given that hydrogen peroxide alone can mediate some sulfur oxidation reactions (such as, *J. Am. Chem. Soc.* 2012, 134, 10765–10768), the inclusion of essential negative control experiments (without the enzyme, and with liquid secretate lacking the target enzyme) .
5. The total turnover number (TTN) is a key metric for assessing the industrial practicality and scalability. While the manuscript highlights the advantages of UPOs, providing experimental TTN data for representative transformations would significantly bolster the claim of their utility for potential industrial applications.
6. To ensure reproducibility and reader accessibility, it is recommended that key experimental details for the biocatalysts should be included directly in the Supporting Information of this work, rather than relying solely on multi-layered references. Such experimental details should at least include the gene sequences for rAaeUPO-PaDa-I-H and artUPO, the detailed enzyme production and purification protocols, and methods for determining enzyme concentration and activity.
7. To further underscore the synthetic utility and potential impact of this new methodology, it would be highly compelling to demonstrate its application in the synthesis of enantioenriched stereogenic-at-sulfur(IV) compounds or their key intermediates listed in Figure 1a, such as the agrochemical Suffoxaflor (1d).
8. Various other issues to be addressed:
 - 8.1. In the abstract, the terms sulfilimine and sulfenimine appear to be inadvertently transposed when describing the substrate-product relationships for sulfoximine and sulfinimine generation.
 - 8.2. The statement in the introduction regarding the catalysis of sulfide-to-sulfoxide transformations by various oxygenases (e.g., P450s, BVMOs) should be supported by appropriate citations.
 - 8.3. The figure labels in the introduction ("Fig 1d" and "Fig 1e") should be formatted consistently as "Figure 1d" and "Figure 1e" throughout the text to adhere to standard academic style.
 - 8.4. The enantiomeric excess (ee) values for the recovered sulfilimines (S)-4k, (S)-4o, and (S)-4s are missing from Table 1 and should be provided.
 - 8.5. In the Methods section, the volume of KPi buffer is stated redundantly as "10 mL" twice ("KPi buffer (10 mL... 10 mL)"); this should be corrected.
 - 8.6. The page range for Reference 8 is incorrect, and the correct citation should be *Nat. Chem.* 16, 1301–1311 (2024).

Reviewer #3

(Remarks to the Author)

The manuscript by Li et al. reports the enzymatic oxygenation of sulfilimines and sulfenimines that yields enantiomerically enriched sulfoximines and sulfinimines, respectively, on preparative scale, using two unspecific peroxygenases (UPOs), previously described, namely the rAaeUPO-Pada-I-H and the artUPO. The study addresses an interesting and new biosynthetic method to obtain stereogenic-at-sulfur(IV) compounds based on oxygenation by two known UPOs. Overall, this work represents a novel application of UPO catalysis that merits publication. However, several corrections should be addressed before being accepted:

1. Our main concern refers to the oxidation state of sulfur (S) considered in the different compounds throughout the manuscript. The authors have mixed the "formal method" (electronegativity-based assignment) and the "functional method" (conceptual & empirical rules), creating inconsistencies. They should use the same criteria within the manuscript for all sulfur compounds. Some examples of this:

- Line 30: in our opinion sulfoxides (like Omeprazole) has a (+2) oxidation state. If omeprazole would have a (+4) oxidation state, as depicted in Figure 1a, the other organosulfur compounds (in Figure 1) such as sulfenamides and sulfoximines, should have a higher oxidation state.

- Line 64: If with "the S(IV) oxidation reactions" the authors mean that the substrates are S(IV), this assumption is not correct. The authors should correct this sentence and write instead, "the S(IV) oxidation products". The authors should revise and correct the oxidation state of S accordingly throughout the manuscript.

2. The authors mention in several parts of the manuscript (lines 22, 69, 83...) that the sulfinimines series are generated by a selective "stereodivergent" synthesis. Indeed, in our opinion the term "stereodivergent" cannot be strictly applied as authors do in these cases, since the substrate used for the synthesis of (R) and (S) sulfinimines is not the same (e.g. in Figure 1e in one case R1 is H, and in the other is CH₃). The authors should clarify this "not strict" use of the "stereodivergent" term in the manuscript.

Moreover, there are some cases in that the reaction, in principle, could be strictly stereodivergent because the substrate is the same (in the case of products 7a, 9a, 12a and 13a), although, only 12a is really a stereodivergent synthesis. Therefore, we would consider these reactions (except that of 12a) as stereoselective, enantioselective, or asymmetric synthesis.

Lines 225-227: We agree with the authors in the inability of rAaeUPO-PaDa-I-H to convert bulky substrates due to its narrow pocket. However, artUPO should work with both bulky and small substrates. Why authors did not test artUPO with both (isopropyl and tert-butyl) series? Indeed, this would provide a strict stereodivergence.

3. Lines 129-132: The authors mention that rAae-UPO-Pada-I-H was not selected to be tested with N-cyanosulfilimines because of its much lower conversion when used in previous experiments. However, in those experiments, the authors only tested 2 N-cyanosulfilimines. In the case of one of these two (entry 1), the conversions attained by both UPOs are really low and similar. We think that the authors should have tested more substrates before excluding the rAae-UPO-Pada-I-H of this series. In our opinion, more reactions should be done with rAae-UPO-Pada-I-H and the results included in the manuscript.

4. Lines 181-187: The authors explain (based on docking experiments) that the high enantioselectivity attained with the aromatic substrate 4e can be due to a hydrophobic interaction between the benzene ring and several hydrophobic aminoacids of artUPO. However, comparing all the results in Table 1A), it can be observed that the substrates with para-substituted rings (e.g. 4b and 4d) show better conversion and enantioselectivity than the meta counterparts. The authors should compare a para-substituted and a meta-substituted substrate to further support their hypothesis.

5. There is continued confusion in the text regarding the terminology of sulfur compounds:

- Line 20: Write "In the sulfilimine series," instead of "In the sulfenimine series,"

- Line 21: Write "In the sulfenimine series," instead of "In the sulfilimine series,"

- Line 45: Write "chiral sulfinamides" instead of "chiral sulfonamides"

- Line 70: Write "sulfinimine" instead of "sulfilimine"

- Line 79: Write "sulfinamides" instead of "sulfonamides"

- Line 203: Write "Ellman's sulfinamide" instead of "Ellman's sulfonamide"

- In Figure 1d: Write and "(S)-5-sulfoximine," instead of "(S)-5"

- In Figure 1e: Write "(S)-sulfinimine" instead "of (S)-sulfinime" and "(R)-sulfinimine" instead "of (R)-sulfinime".

6. Other concern refers to references in the manuscript. There are several aspects to be corrected, as outlined below.

- Reference 6 is repeated. Indeed, it is already included in reference 1 (1b). Please, correct all references accordingly in the text (lines 37, 39) and in the References section.

- Reference 28, authors mentioned that sulfinimines are used as synthetic intermediates but this paper refers to sulfinylamines.

- Line 53: Please, cite papers of sulfoxidations by UPOs instead of references 11 and 12 that do not refer to this particular reaction.

- Line 79 (Figure 1b), reference 8 is denoted as "Tian, Xie, Guo and coworkers", however, reference 8 in the References section is written as "Wei, Wang, Tian...". Please, write it correctly in legend for Figure 1. The same happens with reference 16 (line 81).

7. Other corrections:

- Lines 60-63: these lines, which seem to refer to Figure 1c (although it is not cited in the text), are a bit confusing because they are referring to an artUPO sulfoxidation reaction in a previous work of the authors (compared to AaeUPO) but they only show the reaction by AaeUPO and not the one by artUPO (in Figure 1c) and moreover this figure is not cited in the text. Please, include the reactions of both UPOs in Figure 1c.

- Line 110: delete "box"?

- Table 1: some corrections are necessary. Sometimes the overall yield does not match with the sulfilimine recovered and the sulfoximine produced. Please, revise all the data in this Table.

- Table 2: the nomenclature for the boxes 2A and 2B is confusing. Both are included in section A) but they are not named 1A and 2A. Also, in section D), (S)-13i, write "400 mg" instead "400.

- Methods section: please, provide the concentration of buffers and express all concentrations in molar concentration (e.g. mM, etc.).

Reviewer #4

(Remarks to the Author)

Version 1:

Reviewer comments:

Reviewer #1

(Remarks to the Author)

The authors have adequately addressed my concerns in the second round, now it think it is ready to publish as is.

Reviewer #2

(Remarks to the Author)

Recommendation: Publish in Nature Communications after minor revisions.

The revisions and additional experiments have significantly strengthened the manuscript and addressed the majority of my concerns. Below are my remaining suggestions for further improvement:

1. The manuscript would benefit from explicitly stating the turnover number (TTN) values in the main text, rather than only in the Supporting Information (SI).
2. The labels "Figure 1C" in lines 67-68 should be formatted in lowercase, as "Figure 1c"
3. The compound numbering in Table 2C is inconsistent and non-sequential. Please ensure that (S)-13j on line 291 is changed to (S)-13i to align with the table, and address the gap in the table's numbering where (S)-13j is missing.

Reviewer #3

(Remarks to the Author)

In the revised manuscript the authors have addressed all the corrections suggested for the earlier version. Therefore, in our opinion it can be accepted for publication in Nature Communications

Reviewer #4

(Remarks to the Author)

RESPONSE TO REVIEWER COMMENTS

Reviewer #1

Li and colleagues designed a biocatalytic method for converting sulfur-containing substrates into chiral sulfoxides using unspecific peroxygenases. Specifically, based on the selection of different peroxygenase enzymes, biocatalytic oxygenation of sulfilimines and sulfenimines to form enantiomerically enriched sulfoximines and sulfinimines on preparative scale was achieved. The sulfoximines are generated in up to 99% ee via kinetic resolution reaction using ArtUPO, while the sulfinimines were obtained with complementary chirality using the combination of ArtUPO and AaeUPO. These two methods indeed provide new catalytic insights into the unspecific peroxygenase catalysis, and also establish new catalytic pathways for synthesizing chiral sulfur-compounds. I found this work interesting and promising, and well-designed.

Response: we thank reviewer #1 for their positive assessment of our work, and also for their insightful and useful comments that follow.

Prior to publishing the manuscript, some corrections are necessary to improve the work further. Details are shown below.

1. In the introduction section, it was too focused on the sulfur-containing compounds (e.g., their importance, synthesis, etc.). To broaden the interest of readership, it would be great if the authors could share some attention to the peroxygenase catalysis, particularly considering the fact that the advances of this class of enzymes are expanding fast, lots of new catalytic pathways and new products have been reported.

Response: We certainly share the reviewer's view that peroxygenase catalysis is an important and rapidly expanding field and are glad to have opportunity to highlight this. We have added more text to the introduction and relevant citations (17-28, 30-31) in response to the Reviewer's suggestion.

2. The authors state: "However, to the best of our knowledge, all published syntheses of enantioenriched sulfoximines from sulfilimines start from enantioenriched sulfilimine precursors; as sulfilimines are themselves chiral and configurationally stable, stereochemical information in the precursor is retained following oxidation". To enhance the clarity of this manuscript for readers, could the authors consider providing a summary table or a concise listing of these reported synthesis methods in the Supporting Information?

Response: We thank the reviewer for this useful point – we completely agree that the wording of the statement in our original submission is not as clear as it might be. We think that this can be rectified by replacing the sentence highlighted with a new sentence that we think is both clearer, and more descriptive in terms of the chemistry. Thus we have replaced the text reviewer #1 noted with:

'However, to the best of our knowledge, all published syntheses of enantioenriched sulfoximines from sulfilimines rely on the stereospecific oxidation of an enantiomerically enriched sulfilimine precursors,^{7a} with the requisite sulfilimine starting materials typically prepared from sulfides, e.g. using an asymmetric transition metal catalysed nitrene transfer reaction.^{37,38}

To further clarify how both the asymmetric nitrene transfer and stereospecific oxidation steps operate, 2 new citations (37 and 38) have been added that each feature both steps: Wang, J., Frings, M. & Bolm, C. Enantioselective Nitrene Transfer to Sulfides Catalyzed by a Chiral Iron Complex. *Angew. Chem. Int. Ed.* 52, 8661–8665 (2013); Lebel, H., Piras, H. & Bartholoméüs. Rhodium-Catalyzed Stereoselective Amination of Thioethers with

N-Mesyloxycarbamates: DMAP and Bis(DMAP)CH₂Cl₂ as Key Additives. *Angew. Chem. Int. Ed.* 53, 7300–7304 (2014).

3. The description of enzymes in Tables 1 and 2 is inaccurate and lacks clarity. Please specify the amount of enzyme used in each reaction. (for example: table 1: artUPO secretate (1 mL); table 2: rAaeUPO-PaDa-I-H or art UPO and 6 (0.2 mmol))

Response: done – the amount of enzyme and their activity were added to the captions for both Tables 1 and 2.

4. The overall yield for (S)-4i+(S)-5i appears to be incorrect based on the data provided.

Response: we apologize for this error. The data were re-checked, and the overall yield of 80% is correct, but the mistake was in the reported yield for (S)-4i. This has been corrected to 44% in Table 1 of the revised manuscript, meaning that the overall yield is now logical and correct.

5. The authors utilize rAaeUPO-PaDa-I-H (Class I) and ArtUPO (Class II) as representative enzymes and demonstrate differing reactivity and selectivity patterns. To further broaden the impact of this study, have the authors considered exploring the applicability of these findings to other UPO sources? Specifically, do other Class I and Class II UPOs exhibit similar reactivity and selectivity patterns?

Response: In response to this suggestion, we have challenged substrates 4a, 6a-*i*Pr and 6a-*t*Bu with the Class I *Dca*UPO from *Daldinia caldariorum* and Class II *Cma*UPO from *Coprinopsis marcescibilis*. Interestingly, the (S)- and (R)- selectivities for substrates 6 exhibited by Class I artUPO and Class II rAaeUPO-PaDa-I-H are conserved in enzymes from the same Class. We have added clear mention of these results to the manuscript, with addition detail in the SI (sections 2.6 and 2.9). Details for the cloning and expression of these two enzymes have also been added to the SI, in Section 4.

6. Please clearly explain the meanings of the conversions (Table 1 and Table 2) and how they were determined.

Response: as all of the UPO transformations in Tables 1 and 2 were performed on preparative scale, the % yields in both Tables are not conversions - rather they are true, isolated yields, based on pure, isolated sample of the product (or both products in Table 1) obtained following column chromatography (see SI). To make this positive feature of the work clearer, we have added the following additional note to the captions for both Tables 1 and 2: 'All % yields refer to pure product, isolated by column chromatography, in a preparative scale biotransformation'

7. I would prefer to see a reaction time course of both model reactions to see the initial reaction rate and robustness of the overall reaction scheme. It would be great if the authors can provide this and place them in the Supplementary Information.

Response: These have been added to the revised SI (see section 2.5, 2.9. 2.10).

8. Line 101: Zhang, Zhang and coworkers was wrong.

Response: we had originally highlighted the names of the two corresponding authors for this study (L. Zhang and X. Zhang). Assuming the reviewer is asking that we change to the first author instead, this is done – changed to 'Wang and coworkers'. For consistency, we have also changed 'Arnold and coworkers' to 'Farwell and coworkers' earlier in the same sentence.

9. Supporting Information, page 68: Please describe in detail the method used to determine the UPO concentration (e.g., Bradford assay, spectrophotometric method) and/or enzyme

activity. It is essential for reproducibility that detailed information regarding the enzyme activity (specific activity, total units used per reaction) and enzyme preparation (source, purification stage, formulation buffer) are provided both in the main text (Experimental Section) and comprehensively in the Supplementary Information.

Response: The requested detail has been added to the updated Supplementary Information, at the end of section 4.

Reviewer #2 (Remarks to the Author):

This work presents an innovative biocatalytic oxidation strategy for the asymmetric synthesis of valuable stereogenic-at-sulfur compounds. By harnessing the power of unspecific peroxygenases (UPOs), the authors have successfully developed a kinetic resolution process for sulfilimines to access sulfoximines with up to 99% ee, and a stereodivergent oxygenation of sulfenimines to furnish either enantiomer of sulfenimines with high enantioselectivity. The substrate scope is impressively broad, encompassing a diverse range of sulfilimine and sulfenimine derivatives with generally good yields and high enantioselectivities. Given that such efficient and selective biocatalytic methods for constructing these important chiral sulfur(IV) centers were previously unknown, this study represents a landmark contribution that significantly enriches the synthetic toolbox for asymmetric sulfur chemistry and expands the synthetic applicability of UPOs. The work is therefore suitable for publication in Nature Communications following appropriate revisions to address the points outlined below.

Response: we thank reviewer #2 for their positive assessment of our work, and also for their insightful and useful comments that follow.

1. The introduction provides a compelling overview of the importance of stereogenic sulfur(IV) compounds. To further strengthen the manuscript, the organization and literature coverage should be enhanced. For example, the extensive background on chemical methods for sulfoximine synthesis in the first paragraph of the Results and Discussion would be more appropriately placed in the introduction.

Response: regarding the placement of the discussion on background to sulfoximine synthesis, we respectfully ask that it remains where it was as placed in the original submission. We agree that such background would normally feature in the introduction, and indeed we did consider this when writing the manuscript. However, because this manuscript features two distinct studies towards different S(IV) functional groups, from different starting materials, we think that introducing the background to each in turn is preferable. Placing all the literature discussion at the start would lead to the introduction being very long, and we think that differentiating between the different methods and functional groups (which tend to have very similar nomenclature) would be more difficult for the reader to follow.

Furthermore, while kinetic resolution and organocatalysis are mentioned, the introduction would benefit from acknowledging other catalytic strategies, such as transition metal catalysis (e.g., J. Am. Chem. Soc. 2025, 147, 2137–2147; ACS Catal. 2025, 15, 5511–5530, Nat. Commun. 2025, 16, 2310; JACS Au 2023, 3, 700–714). Additionally, presenting 'kinetic resolution' and 'organocatalysis' as parallel categories is conceptually imprecise.

Response: thanks for the suggestions. We note that two of the studies noted were already cited in the original manuscript: ACS Catal. 2025, 15, 5511–5530 (reference 7g) and JACS Au 2023, 3, 700–714 (reference 7a).

We fully agree that kinetic resolution and organocatalysis are not parallel strategies. We did not intend to imply this and apologize that our original wording led to this impression. Several of the references in citations 7a-g feature both strategies (e.g.

review articles), hence we would like to retain mention of both strategies, but to avoid conflating these concepts, we have altered the wording in this section, which now reads: ‘often based on kinetic resolution strategies, transition metal catalysis, or methods using organocatalysts’

Notably, we added the phrase ‘transition metal catalysis’ to this description also, and added citation to the other 2 previous studies (J. Am. Chem. Soc. 2025, 147, 2137–2147 and Nat. Commun. 2025, 16, 2310) mentioned by the reviewer, as new references 7h and 7i.

2. For the model substrate rac-4 in the sulfilimine series, the initial results, while promising, were moderate ((S)-4a: 35% yield, 32% ee; (S)-5a: 22% yield, 74% ee). It is unclear whether systematic reaction optimization—such as varying enzyme loading, H₂O₂ concentration, temperature, or reaction time—was performed for this model reaction. Including these details or comments on optimization efforts in the Supporting Information would be valuable.

Response: Surprisingly little optimisation was needed to optimise the kinetic resolution reactions in the sulfilimine series. The optimisation that was done is included in the SI, in section 2.6. Here, experiments using different sulfilimines (including replacing the CN group) and H₂O₂ were explored and reaction conversion reported, using both AaeUPO and artUPO. Enzyme loading, temperature and the organic co-solvent were not explored; our groups have broad experience developing preparative scale UPO transformations, and the combination of pH 7 buffer, 20% CH₃CN and RT have been proven to work well in a range of transformations, and we were fortunate that this was also the case in this study. In addition to the UPO used, the amount of H₂O₂ added tends to have the largest impact on UPO transformations. For these reactions, the H₂O₂ loading was dictated by us striving to reach ≈50% conversion for optimal kinetic resolution, hence relatively little deviation from 0.6 equivalents (the standard method) was explored, although small variations that led to improvements were tested during scale-up work, as detailed in section 2.5 of the SI. We acknowledge that these SI details were not signposted in the manuscript, hence have added the following direction to the revised manuscript when the reaction conditions are introduced: ‘(for optimisation details, see SI sections 2.5 and 2.6)’.

3. The assignment of absolute configuration for certain compounds requires further verification. The determination for (S)-5a, based on comparison of optical rotation with a literature value obtained for a sample of significantly higher enantiopurity (74% ee vs. 99% ee), could be strengthened. Synthesizing an authentic reference standard for direct comparison via chiral HPLC would provide more definitive proof.

Response: We welcome the opportunity to further strengthen the stereochemical assignments in the study. To address this comment, we did as the reviewer suggested and synthesized an enantioenriched sample of (S)-5a. This was done using a known chemical synthesis and resolution approach [described in: Brandt, J.; Gais, H.-J. An efficient resolution of (-)-S-methyl-S-phenylsulfoximine with (+)-10-camphorsulfonic acid by the method of half-quantities. Tetrahedron. 8, 902–912 (1997)] cited in the revised SI as reference 8b. This method allowed for the chemical synthesis of (S)-5 enriched to 84% ee, which was analysed by the same chiral HPLC method used to determine the ee of our UPO generated sample of (S)-5a and confirm that the same (S)-isomer was the major enantiomer in both samples. Full synthetic details and HPLC data for the synthetic sample have been added to the revised SI on pages S32–33.

Similarly, the assignment of (R)-12a also relies on comparison of optical rotation with literature data (ref. 30). However, the cited reference does not contain the relevant data for (R)-12a.

Verification of this citation or additional evidence for configuration assignment is recommended.

Response: We sincerely apologize that reference 30 was an incorrect citation. In the revision manuscript this has been corrected to: 'Roe, C., Hobbs, H. & Stockman, R. A. Multicomponent Synthesis of Chiral Sulfinimines. Chem. Eur. J. 17, 2704–2708 (2011).' Note that due to other changes, this reference has become reference 49 in the revision. The optical rotation data for (R)-12a and the same compound in Stockman and coworkers' study both have a high negative value, thus providing strong evidence for the assigned absolute stereochemistry of (R)-12a

4. The use of enzyme secretate is practical, but it raises the question of whether other components in the secretate could contribute to or interfere the catalysis. Additionally, given that hydrogen peroxide alone can mediate some sulfur oxidation reactions (such as, J. Am. Chem. Soc. 2012, 134, 10765–10768), the inclusion of essential negative control experiments (without the enzyme, and with liquid secretate lacking the target enzyme).

Response: Enzyme-free control reactions have been added to the SI (see SI, Sections 2.6 and 2.9.2) for each reaction system. We are not in possession of a secretate that does not contain a UPO. However, we maintain that the marked opposite enantioselectivity exhibited by secretates containing either Class I or Class II UPOs constitute overwhelming evidence of their respective contribution to reaction products. The major contaminant of *Pichia* secretates is likely to be the flavin dependent alcohol oxidase (AOX), for which we can find no record in the literature of a capacity for sulfoxidation, asymmetric or otherwise.

5. The total turnover number (TTN) is a key metric for assessing the industrial practicality and scalability. While the manuscript highlights the advantages of UPOs, providing experimental TTN data for representative transformations would significantly bolster the claim of their utility for potential industrial applications.

Response: TTNs for representative biotransformations have been added to the SI with the time courses in Figures S2 and S3. These values of approximately 10^3 to 10^4 are within the range of figures reported for UPOs previously working with this amount of substrate.

6. To ensure reproducibility and reader accessibility, it is recommended that key experimental details for the biocatalysts should be included directly in the Supporting Information of this work, rather than relying solely on multi-layered references. Such experimental details should at least include the gene sequences for rAaeUPO-PaDa-I-H and artUPO, the detailed enzyme production and purification protocols, and methods for determining enzyme concentration and activity.

Response: The requested detail has been added to the updated Supplementary Information, Section 4.

7. To further underscore the synthetic utility and potential impact of this new methodology, it would be highly compelling to demonstrate its application in the synthesis of enantioenriched stereogenic-at-sulfur(IV) compounds or their key intermediates listed in Figure 1a, such as the agrochemical Sulfoxaflor (1d).

Response: this is an excellent suggestion, and one that we also considered and tested. artUPO can perform the sulfilimine to sulfoximine oxidation needed to make Sulfoxaflor. However, the additional stereogenic center (the benzylic methyl group) and facile epimerization of both this stereocenter, and the sulfoximine at RT, mean that neither the *dr* or *ee* can be reliably measured. It is notable that commercial Sulfoxaflor

is sold as a roughly 1:1 mixture of diastereoisomers, in racemic form (see https://www.fao.org/fileadmin/templates/agphome/documents/Pests_Pesticides/JMPR/Evaluation11/Sulfoxaflor.pdf for details). In our hands, when we prepared the requisite sulfilimine to test this idea, we were unable to obtain it as a single diastereoisomer, and following UPO oxidation of this mixture, *dr* and *ee* measurements of the oxidation products were inconsistent and changed over time. This is likely due to epimerization of either/both stereocenters. Therefore, we chose not to report these results as they cannot be easily reproduced. Indeed, given its proclivity to epimerize at RT, it could be argued that there is little practical value in developing an enantioselective Sulfoxaflor synthesis.

Instead, we decided to focus on alternative applications that showcase the synthetic utility of UPOs, specifically the 2 examples at the end of Table 2.

8. Various other issues to be addressed:

8.1. In the abstract, the terms sulfilimine and sulfenimine appear to be inadvertently transposed when describing the substrate-product relationships for sulfoximine and sulfinimine generation.

Response: many thanks reviewer #2 for spotting this mistake – corrected.

8.2. The statement in the introduction regarding the catalysis of sulfide-to-sulfoxide transformations by various oxygenases (e.g., P450s, BVMOs) should be supported by appropriate citations.

Response: Reference 9 (in the original, and in the revision) relates to a review on this topic, and to better signpost this, we have added '9' a second time directly after 'the conversion of simple sulfides to sulfoxides' in this section, which we hope is clearer. Moreover, we have also added some additional sulfoxidation references for P450s and BVMOs (references 10-13).

8.3. The figure labels in the introduction ("Fig 1d" and "Fig 1e") should be formatted consistently as "Figure 1d" and "Figure 1e" throughout the text to adhere to standard academic style.

Response: done

8.4. The enantiomeric excess (*ee*) values for the recovered sulfilimines (S)-4k, (S)-4o, and (S)-4s are missing from Table 1 and should be provided.

Response: In most cases, we were unable to measure the *ee* of the sulfilimines (S)-4 directly using chiral HPLC, due to challenging chiral HPLC separation. Rather it was necessary to oxidize the sulfilimine to the corresponding sulfonimine (R)-5 and measure the *ee* at this stage, following stereospecific oxidation using *m*-CPBA; this was noted in the original manuscript, with full details in the SI. For the three cases noted by the reviewer, we were unable to achieve satisfactory chiral HPLC separation of the sulfilimines (S)-4, and the *m*-CPBA oxidations failed, hence why *ee* data were not recorded for these 3 substrates.

8.5. In the Methods section, the volume of KPi buffer is stated redundantly as "10 mL" twice ("KPi buffer (10 mL... 10 mL)"); this should be corrected.

Response: corrected

8.6. The page range for Reference 8 is incorrect, and the correct citation should be Nat. Chem. 16, 1301–1311 (2024).

Response: corrected

Reviewer #3 (Remarks to the Author):

The manuscript by Li et al. reports the enzymatic oxygenation of sulfilimines and sulfenimines that yields enantiomerically enriched sulfoximines and sulfinimines, respectively, on preparative scale, using two unspecific peroxygenases (UPOs), previously described, namely the rAaeUPO-Pada-I-H and the artUPO.

The study addresses an interesting and new biosynthetic method to obtain stereogenic-at-sulfur(IV) compounds based on oxygenation by two known UPOs. Overall, this work represents a novel application of UPO catalysis that merits publication.

Response: we thank reviewer #3 for their positive assessment of our work, and also for their insightful and useful comments that follow.

However, several corrections should be addressed before being accepted:

1. Our main concern refers to the oxidation state of sulfur (S) considered in the different compounds throughout the manuscript. The authors have mixed the “formal method” (electronegativity-based assignment) and the “functional method” (conceptual & empirical rules), creating inconsistencies. They should use the same criteria within the manuscript for all sulfur compounds. Some examples of this:

- Line 30: in our opinion sulfoxides (like Omeprazole) has a (+2) oxidation state. If omeprazole would have a (+4) oxidation state, as depicted in Figure 1a, the other organosulfur compounds (in Figure 1) such as sulfinamides and sulfoximines, should have a higher oxidation state.

- Line 64: If with "the S(IV) oxidation reactions" the authors mean that the substrates are S(IV), this assumption is not correct. The authors should correct this sentence and write instead, “the S(IV) oxidation products”.

The authors should revise and correct the oxidation state of S accordingly throughout the manuscript.

Response: we absolutely agree with the reviewer. We have made all of the corrections specified, and several others based on the same systematic errors (see highlighted revised manuscript). In most instances, the inconsistencies were fixed simply by removing incorrect uses of '(IV)' from the compound descriptors. Notwithstanding the errors, noting the oxidation state as frequently as was done was arguably not necessary anyway.

2. The authors mention in several parts of the manuscript (lines 22, 69, 83...) that the sulfinimines series are generated by a selective “stereodivergent” synthesis. Indeed, in our opinion the term “stereodivergent” cannot be strictly applied as authors do in these cases, since the substrate used for the synthesis of (R) and (S) sulfinimines is not the same (e.g. in Figure 1e in one case R1 is H, and in the other is CH₃). The authors should clarify this “not strict” use of the “sterodivergent” term in the manuscript. Moreover, there are some cases in that the reaction, in principle, could be strictly stereodivergent because the substrate is the same (in the case of products 7a, 9a, 12a and 13a), although, only 12a is really a stereodivergent synthesis. Therefore, we would consider these reactions (except that of 12a) as stereoselective, enantioselective, or asymmetric synthesis.

Response: Again, we completely agree with the reviewer – thank you reviewer #3 for holding to account on these important points of precision! We considered all usages of the term “stereodivergent” in the manuscript, and decided that none of necessary, and hence in all cases they were either removed, or replaced with a simpler description (e.g. ‘(R)- and (S)-sulfinimine products’ in the abstract).

Lines 225-227: We agree with the authors in the inability of rAaeUPO-PaDa-I-H to convert bulky substrates due to its narrow pocket. However, artUPO should work with both bulky and small substrates. Why authors did not test artUPO with both (isopropyl and tert-butyl) series? Indeed, this would provide a strict stereodivergence.

Response: We included details of artUPO transformation with both bulky and small substrates in the original submission; the results are summarised in Box 2A of Table 2. As the reviewer predicted, artUPO does indeed convert both, but for the smaller substrates (e.g. to make 7a and 9a) the enantioselectivity was low. This is common for artUPO – compared with AaeUPO, it often is able to promote higher conversions, but at the cost of lower selectivity. Box 2A also includes the formation of (S)-12a from an isopropyl precursor, thus providing the strict stereodivergence noted by the review in this one case.

3. Lines 129-132: The authors mention that rAae-UPO-Pada-I-H was not selected to be tested with N-cyanosulfilimines because of its much lower conversion when used in previous experiments. However, in those experiments, the authors only tested 2 N-cyanosulfilimines. In the case of one of these two (entry 1), the conversions attained by both UPOs are really low and similar. We think that the authors should have tested more substrates before excluding the rAae-UPO-Pada-I-H of this series. In our opinion, more reactions should be done with rAae-UPO-Pada-I-H and the results included in the manuscript.

Response: For the two N-cyanosulfilimines noted, conversion was 3.5x higher (entry 1) and 5.9 x higher (entry 3, see SI section 2.6) for artUPO compared with AaeUPO, which we feel is a relatively strong steer. Although we agree this is a small sample size. Therefore, we later tested two more N-cyanosulfilimines (4l and 4m) using AaeUPO and 0% conversion was observed in both cases, further indicating that artUPO is superior for this system. We neglected to mention this in the original submission, therefore, have added the following note to Section 2.6 of the SI: ‘Sulfilimines 4l and 4m were also tested using rAaeUPO-PaDa-I under the conditions used above, with 0% conversion observed, further validating our decision to prioritise artUPO for this substrate class.’

4. Lines 181-187: The authors explain (based on docking experiments) that the high enantioselectivity attained with the aromatic substrate 4e can be due to a hydrophobic interaction between the benzene ring and several hydrophobic aminoacids of artUPO. However, comparing all the results in Table 1A), it can be observed that the substrates with para-substituted rings (e.g. 4b and 4d) show better conversion and enantioselectivity than the meta counterparts. The authors should compare a para-substituted and a meta-substituted substrate to further support their hypothesis.

Response: The yields of the transformation of the para-methoxy substrate 4b and meta-methoxy substrate 4c – the only substrates for which the requested comparison can be made directly – are 41% and 29% respectively. The *E* values (relating conversion to enantioselectivity) for 4b and 4c are 42 and 27. Both are arguably within the same range. However, there is a difference, and we recognize the reviewer’s request for more detail. The model suggests that while a *para* substituent would be accommodated well within the hydrophobic pocket described, a *meta*-substituent may clash with side-chains of

Val69 or Ile91, disfavouring binding to some degree. A line has been added to the manuscript to highlight this.

5. There is continued confusion in the text regarding the terminology of sulfur compounds:

- Line 20: Write “In the sulfilimine series,” instead of “In the sulfenimine series,”

Response: corrected

- Line 21: Write “In the sulfenimine series,” instead of “In the sulfilimine series,”

Response: corrected

- Line 45: Write “chiral sulfinamides” instead of “chiral sulfonamides”

Response: corrected

- Line 70: Write “sulfinimine” instead of “sulfilimine”

Response: corrected

- Line 79: Write “sulfinamides” instead of “sulfonamides”

Response: corrected

- Line 203: Write “Ellman’s sulfinamide” instead of “Ellman’s sulfonamide”

Response: corrected

- In Figure 1d: Write and “(S)-5-sulfoximine,” instead of “(S)-5”

Response: corrected

- In Figure 1e: Write “(S)-sulfinimine” instead “of (S)-sulfinime” and “(R)-sulfinimine” instead “of (R)-sulfinime”.

Response: corrected

6. Other concern refers to references in the manuscript. There are several aspects to be corrected, as outlined below.

- Reference 6 is repeated. Indeed, it is already included in reference 1 (1b). Please, correct all references accordingly in the text (lines 37, 39) and in the References section.

Response: corrected – as the other reviewers also requested additions to the introductory sections, several new citations have been added, and hence many of the references have been renumbered.

- Reference 28, authors mentioned that sulfinimines are used as synthetic intermediates but this paper refers to sulfinylamines.

Response: yes, this is correct. We have replaced the previous reference 28 with the following review article that is much more relevant: Zhou, P., Chen, B.-C., Davis, F. A. Recent advances in asymmetric reactions using sulfinimines (N-sulfinyl imines). *Tetrahedron*, 60, 8003–8030 (reference 47 in the revision)

- Line 53: Please, cite papers of sulfoxidations by UPOs instead of references 11 and 12 that do not refer to this particular reaction.

Response: In response to a related comment by reviewer 2, changes were made to better signpost reference 9, a review cited in the original. We have also cited references sulfoxidation papers (references 15 and 16) in this section of text.

- Line 79 (Figure 1b), reference 8 is denoted as “Tian, Xie, Guo and coworkers”, however, reference 8 in the References section is written as “Wei, Wang, Tian...”. Please, write it correctly in legend for Figure 1. The same happens with reference 16 (line 81).

Response: corrected – in the original, both were written highlighting the names of the corresponding authors, but we have changed to the first author in both instances instead.

7. Other corrections:

- Lines 60-63: these lines, which seem to refer to Figure 1c (although it is not cited in the text), are a bit confusing because they are referring to an artUPO sulfoxidation reaction in a previous work of the authors (compared to AaeUPO) but they only show the reaction by AaeUPO and not the one by artUPO (in Figure 1c) and moreover this figure is not cited in the text. Please, include the reactions of both UPOs in Figure 1c.

Response: we completely agree that this section was not clear enough. To rectify this, we have modified Figure 1C to show both UPO reactions, added direct reference to Figure 1C to the main text, and modified the text describing both UPO reactions significantly (see highlighted manuscript). We think the section is much clearer as a result.

- Line 110: delete “box”?

Response: done

- Table 1: some corrections are necessary. Sometimes the overall yield does not match with the sulfilimine recovered and the sulfoximine produced. Please, revise all the data in this Table.

Response: the same point was raised by reviewer #1. We have checked carefully all entries in Table 1 and corrected all in which correction was needed.

- Table 2: the nomenclature for the boxes 2A and 2B is confusing. Both are included in section A) but they are not named 1A and 2A.

Response: We agree. We have changed to 1A and 2A as suggested and updated all references to them accordingly.

Also, in section D), (S)-13i, write “400 mg” instead “400 img”.

Response: done

- Methods section: please, provide the concentration of buffers and express all concentrations in molar concentration (e.g. mM, etc.).

Response: done – the pH 7 buffer was 100 mM concentration in all cases.

Reviewer #4 (Remarks to the Author):

Response: Thanks to reviewer #4 also!

RESPONSE TO REVIEWER COMMENTS

To Reviewers #1, #3 and #4: No further changes were requested.

To Reviewer #2:

1. We have added turnover numbers to the main manuscript.
2. The label 'Figure 1C' has been adjusted
3. Line 291: we have corrected '(S)-13j' into '(S)-13i'. In Table 2C, we would prefer to leave the numbering as it. Leaving out (S)-13j was deliberate, as we wanted the letter prefixes to be consistent for analogous substituents across the products '12' in Table 2B, and also in the starting materials '6'.